# Examining Antimicrobial Resistance in *Escherichia coli*: A Case Study in Central Virginia’s Environment

**DOI:** 10.3390/antibiotics13030223

**Published:** 2024-02-28

**Authors:** Chyer Kim, Allissa Riley, Shobha Sriharan, Theresa Nartea, Eunice Ndegwa, Ramesh Dhakal, Guolu Zheng, Claire Baffaut

**Affiliations:** 1Agricultural Research Station, Virginia State University, 1 Hayden Drive, Petersburg, VA 23806, USA; endegwa@vsu.edu (E.N.); rdhakal@vsu.edu (R.D.); 2Department of Biology, Virginia State University, 1 Hayden Drive, Petersburg, VA 23806, USA; aril0610@students.vsu.edu (A.R.); ssriharan@vsu.edu (S.S.); 3Cooperative Extension, Virginia State University, 1 Hayden Drive, Petersburg, VA 23806, USA; tnartea@vsu.edu; 4Department of Agriculture and Environmental Sciences, Cooperative Research Programs, Lincoln University, 820 Chestnut Street, Jefferson City, MO 65102, USA; zhengg@lincolnu.edu; 5USDA ARS Cropping Systems and Water Quality Research Unit, 241 Agricultural Engineering Building, University of Missouri, Columbia, MO 65211, USA; claire.baffaut@usda.gov

**Keywords:** *E. coli*, AMR, MDR, animal, wastewater, land use water

## Abstract

While environmental factors may contribute to antimicrobial resistance (AMR) in bacteria, many aspects of environmental antibiotic pollution and resistance remain unknown. Furthermore, the level of AMR in *Escherichia coli* is considered a reliable indicator of the selection pressure exerted by antimicrobial use in the environment. This study aimed to assess AMR variance in *E. coli* isolated from diverse environmental samples, such as animal feces and water from wastewater treatment plants (WWTPs) and drainage areas of different land use systems in Central Virginia. In total, 450 *E. coli* isolates obtained between August 2020 and February 2021 were subjected to susceptibility testing against 12 antimicrobial agents approved for clinical use by the U.S. Food and Drug Administration. Approximately 87.8% of the tested isolates were resistant to at least one antimicrobial agent, with 3.1% showing multi-drug resistance. Streptomycin resistance was the most common (73.1%), while susceptibility to chloramphenicol was the highest (97.6%). One isolate obtained from WWTPs exhibited resistance to seven antimicrobials. AMR prevalence was the highest in WWTP isolates, followed by isolates from drainage areas, wild avians, and livestock. Among livestock, horses had the highest AMR prevalence, while cattle had the lowest. No significant AMR difference was found across land use systems. This study identifies potential AMR hotspots, emphasizing the environmental risk for antimicrobial resistant *E. coli*. The findings will aid policymakers and researchers, highlighting knowledge gaps in AMR–environment links. This nationally relevant research offers a scalable AMR model for understanding *E. coli* ecology. Further large-scale research is crucial to confirm the environmental impacts on AMR prevalence in bacteria.

## 1. Introduction

The pervasive use of antibiotics in humans, animals, and agricultural practices has led to the widespread dissemination of antimicrobial-resistant bacteria, posing a significant environmental threat [1,2,3]. Research by Brown et al. [4] identifies partially metabolized antimicrobials from humans and animals as a primary source of environmental contamination. The challenge of AMR is a One Health issue requiring a comprehensive surveillance approach. While the environmental contribution to AMR is speculated, limited research has focused on the environment’s role in AMR [5], and the dimensions of environmental antibiotic pollution and resistance remain unknown [6,7]. Additionally, the absence of quantitative data hampers AMR surveillance efforts, indicating the necessity of a better understanding to reduce resistance spread from the environment to clinical settings [8]. Bacteria from different sources may exhibit distinct resistance patterns due to varying evolutionary trajectories from different selective forces [9], influencing antibiotic treatment decisions. Due to their widespread occurrence in the environment, along with their ability to serve as reservoirs of antibiotic resistance genes, *Escherichia coli* (*E. coli*) is commonly employed as an indicator in numerous antimicrobial resistance monitoring programs [10]. The level of AMR in *E. coli* is also considered a reliable indicator of the selection pressure exerted by antimicrobial use in humans and agricultural practices [10,11]. Notably, AMR elements can be transmitted between and within different ecosystems and bacteria [10].

While the global impact of AMR is a cause for concern, and their national surveillance data remain essential, it is the local data that play a critical role in alerting communities to immediate risks in both environmental and clinical settings [12,13]. Therefore, additional regional surveillance is imperative to enhance our understanding of the emergence and spread of AMR, elucidating the intricate relationship between AMR occurrence and the environment within a “One Health Approach”. This study, serving as a reflective analysis of the regional AMR landscape, aimed to assess the prevalence of AMR in *E. coli* isolated from different types of environmental samples in central Virginia, the location of the corresponding author’s institution. Findings from this study will support the identification of hotspots of antibiotic resistance emergence and dissemination and will be helpful in finding mitigation targets and solutions [6].

## 2. Results and Discussion

The summarized number of *E. coli* isolates obtained per sample type and source and sampling location and the prevalence of their resistance to the 12 antimicrobials tested are presented in Table 1. Due to the limited availability of the same species of livestock and wild avians in different farms and environments, respectively, the number of sampling locations for each sample source acquired was not consistent. For the fecal samples of livestock, only one farm per type of livestock, i.e., horses, sheep, and turkeys, showed an interest in participation in the current study. For the farms with limited availability of livestock, we attempted to obtain three *E. coli* isolates from each of 10 samples (up to 30 *E. coli* isolates) randomly collected at the sampling location. However, due to the lower occurrence of *E. coli* than expected in the horse feces collected, we were unsuccessful in obtaining the target number (30) of *E. coli* isolates, and only 15 isolates were obtained. In addition, due to the lower occurrence of *E. coli* than expected in the goat feces collected, we could obtain 30 *E. coli* isolates only from three sampling locations (farms), consisting of 10 isolates per sampling location. 

The patterns of susceptible, intermediate, and resistant *E. coli* isolates obtained from the feces of the overall livestock are presented in Figure 1A–G. In detail, of the 45 isolates obtained from cattle feces (Figure 1A), resistance to streptomycin (STR) was the most common at 60.0%, followed by resistance to amikacin (AMK, 26.7%), ampicillin (AMP, 8.9%), tobramycin (TOB, 4.4%), and gentamycin (GEN, 9.3%). Approximately 71% (32 isolates) of the isolates tested were resistant to at least one antimicrobial agent (Table 1). Non-susceptibility (either intermediate or resistance) to STR was the most common at 100%, followed by non-susceptibility to AMK (75.6%), TOB (55.5%), GEN (53.3%), and AMP (48.9%). All isolates were non-susceptible to at least one antimicrobial agent (Table 1). The present study revealed that three isolates, one from each farm, were resistant to the antimicrobials tested, while one isolate was non-susceptible to seven antimicrobials. However, none of the isolates displayed MDR, indicating their resistance to less than three antimicrobial categories. The most effective antimicrobials tested were nalidixic acid (NAL) and trimethoprim–sulfamethoxazole (SXT) (Figure 1A).

The prevalence of AMR in the isolates obtained from chicken feces is shown in Figure 1B. Among 45 isolates, resistance to STR was the most common at 68.9%, followed by resistance to AMK (35.6%), AMP (20.0%), and tetracycline (TCY, 15.6%). Approximately 84% (38 isolates) of the isolates tested were resistant to at least one antimicrobial agent (Table 1). Non-susceptibility to STR was the most common at 100%, followed by non-susceptibility to AMK (68.9%), TOB (55.5%), AMP (48.9%), GEN (48.9%), AMC (22.2%), TCY (20.0%), and meropenem (MEM, 13.3%). In other words, all isolates were non-susceptible to at least one antimicrobial agent (Table 1). The present study revealed that one isolate was resistant to five antimicrobials (AMP, AMK, GEN, STR, and TCY) in three categories, indicating MDR, while two isolates were non-susceptible to nine antimicrobials. The most effective antimicrobial was SXT (Figure 1B).

Out of 30 isolates obtained from goat feces (Figure 1C), resistance to AMK was the most common at 53.3%, followed by resistance to STR (33.3%), AMP (20.0%), GEN (13.3%), TCY (13.3%), and TOB (10.0%). Approximately 73% (22 isolates) of the isolates tested were resistant to at least one antimicrobial agent (Table 1). Non-susceptibility to STR was the most common at 96.6%, followed by non-susceptibility to AMK (93.3%), GEN (76.6%), TOB (70.0%), AMP (53.3%), MEM (43.3%), TCY (20.0%), and ciprofloxacin (CIP, 13.3%). All isolates were non-susceptible to at least one antimicrobial agent (Table 1). Although two isolates were resistant and non-susceptible to four and eight antimicrobials, none of them displayed MDR. All isolates obtained from goat feces were susceptible to SXT.

Among 15 isolates obtained from horse feces (Figure 1D), resistance to STR was the most common at 86.7%, followed by resistance to SXT (73.3%), TOB (40.0%), and GEN (20.0%). All isolates were resistant to at least one antimicrobial agent (Table 1). Non-susceptibility to STR and TOB was the most common at 100%, followed by non-susceptibility to GEN (73.3%), SXT (73.3%), AMK (33.3%), AMP (26.7%), MEM (26.7%), and CIP (13.3%). Although six isolates (40%) and one isolate were resistant and non-susceptible to four and eight antimicrobials, respectively, none of the isolates displayed MDR (Table 1). All isolates obtained from horse feces were susceptible to AMC, TCY, and NAL.

Out of 30 isolates obtained from pig feces (Figure 1E), resistance to AMK and STR was the most common at 83.3%, followed by resistance to TCY (26.7%), AMP (23.3%), GEN (13.3%), and TOB (13.3%). Approximately 97% (29 isolates) of the isolates tested were resistant to at least one antimicrobial agent (Table 1). Non-susceptibility to AMK and STR was the most common at 100%, followed by non-susceptibility to TOB (93.3%), GEN (86.6%), AMP (63.3%), amoxicillin–clavulanic acid (AMC, 26.7%), MEM (26.7%), TCY (26.7%), and CIP (13.3%). While three isolates (10%) exhibited MDR, two isolates were non-susceptible to eight antimicrobials. It was also noted that one isolate displayed resistance to six antimicrobials (AMP, AMK, GEN, STR, TOB, and TCY). All isolates obtained from pig feces were susceptible to chloramphenicol (CHL) and SXT.

The prevalence of AMR in 30 isolates obtained from sheep feces, shown in Figure 1F, demonstrated that resistance to STR was the most common at 76.7%, followed by resistance to AMK (60.0%), AMP (46.7%), GEN (6.7%), and TOB (3.3%). Approximately 90% (27 isolates) of the isolates tested were resistant to at least one antimicrobial agent (Table 1). Non-susceptibility to STR was the most common at 100%, followed by non-susceptibility to AMK (96.7%), AMP (90.0%), GEN (80.0%), TOB (73.3%), AMC (23.3%), and MEM (20.0%). Although three isolates were non-susceptible to seven antimicrobials, none of the isolates displayed MDR (Table 1). All isolates obtained from sheep feces were susceptible to TCY, NAL, and SXT.

Out of 30 isolates obtained from turkey feces (Figure 1G), resistance to STR was the most common at 46.7%, followed by resistance to AMK (43.3%), TCY (43.3%), AMP (16.7%), and GEN (13.3%). Twenty-four isolates (80%) were resistant to at least one antimicrobial agent (Table 1). Non-susceptibility to STR was the most common at 96.7%, followed by non-susceptibility to AMK (76.6%), AMP (66.7%), TOB (60.0%), GEN (56.6%), TCY (43.3%), AMC (16.6%), and NAL (13.3%). However, all isolates were non-susceptible to at least one antimicrobial agent (Table 1). While two isolates (6.7%) exhibited MDR, two other isolates showing non-MDR were non-susceptible to seven antimicrobials. One isolate displayed resistance to five antimicrobials (AMP, AMC, AMK, STR, and TCY). All isolates obtained from turkey feces were susceptible to MEM, CHL, and SXT.

Overall, out of the 225 isolates obtained from livestock feces (Figure 1H), resistance to STR was the most common at 63.6%, followed by resistance to AMK (44.4%), AMP (20.0%), and TCY (14.2%). Approximately 83% (187 isolates) of the isolates tested were resistant to at least one antimicrobial agent (Table 1). Non-susceptibility to STR was the most common at 99.2%, followed by non-susceptibility to AMK (80.0%), TOB (68.5%), GEN (65.3%), AMP (57.8%), MEM (17.8%), TCY (16.4%), and AMC (16.0%). All isolates were non-susceptible to at least one antimicrobial agent. Although the prevalence of AMR in the isolates obtained from other livestock demonstrated a similar pattern, the isolates from goat and pig feces revealed higher resistance to AMK. In contrast, the isolates from horse feces showed higher non-susceptibility to TOB than the isolates from other livestock feces. The occurrence of MDR was exhibited in six isolates (2.7%) obtained from chicken (one isolate), pig (three isolates), and turkey (two isolates) only (Table 1). Furthermore, a high percentage (73.3%) of the isolates obtained from horse feces displayed resistance to SXT, while all isolates obtained from other livestock feces were completely susceptible to this antimicrobial agent, indicating animal specificity of the AMR profiles of the isolates. 

Although the types of antimicrobial agents either used or not for livestock in the studied area is unknown, the differences in the AMR profiles of our studied isolates may be due to livestock management differences and to each species of livestock being exposed to different antimicrobials during animal husbandry. A USDA report [15] and several scientists [16,17,18] indicated that the antimicrobial use in agricultural production influences the emergence of AMR. However, regardless of the livestock species tested, resistance and non-susceptibility to STR were the most common, as addressed above. In addition, although all isolates obtained from the livestock feces, except horse feces, were susceptible to SXT, the overall susceptibility of the isolates to CHL was the highest, corresponding to 97.8% (220 isolates), followed by susceptibility to NAL (96.0%), SXT (95.1%), and CIP (92.4%). More importantly, none of the isolates obtained from the livestock feces were susceptible to all antimicrobials tested in this study. 

The prevalence of AMR in the isolates obtained from the feces of wild avian species is presented in Figure 2A–C. Of the 45 isolates obtained from goose feces (Figure 2A), resistance to STR was the most common at 80.0%, followed by resistance to AMK (51.1%), and GEN (11.1%). Approximately 84% (38 isolates) of the isolates were resistant to at least one antimicrobial agent (Table 1). Non-susceptibility to STR was the most common at 100%, followed by non-susceptibility to AMK (95.5%), TOB (84.5%), GEN (82.2%), and AMP (71.1%). Although two and one isolates were resistant and non-susceptible to four and eight antimicrobials, respectively, none of the isolates displayed MDR. All isolates obtained from goose feces were susceptible to CHL and SXT. 

Out of 30 isolates obtained from seagull feces (Figure 2B), resistance to STR was the most common at 76.7%, followed by resistance to AMK (63.3%), AMP (23.3%), TOB (13.3%), and TCY (13.3%). Approximately 93% (28 isolates) of the isolates were resistant to at least one antimicrobial agent (Table 1). Non-susceptibility to STR was the most common at 100%, followed by non-susceptibility to AMK (96.6%), AMP (86.6%), TOB (80.0%), GEN (66.6%), and MEM (50.0%). While two and six isolates exhibited resistance and non-susceptibility to four and seven antimicrobials, only one isolate exhibited MDR. All isolates obtained from seagull feces were susceptible to SXT.

Overall, out of 75 isolates obtained from wild avian feces (Figure 2C), resistance to STR was the most common at 78.7%, followed by resistance to AMK (56.0%), AMP (12.0%), and TOB (10.7%). Sixty-six isolates (88%) were resistant to at least one antimicrobial agent. Non-susceptibility to STR was the most common at 100%, followed by non-susceptibility to AMK (96.0%), TOB (82.7%), AMP (77.3%), GEN (76.0%), MEM (28.0%), and TCY (10.6%). Although the prevalence of AMR in the isolates obtained from wild avian feces demonstrated a similar pattern, the isolates obtained from seagull feces revealed higher resistance to CHL than the isolates obtained from goose feces. 

Among 30 isolates obtained from WWTPs (Figure 3), resistance to STR was the most common at 86.7%, followed by resistance to AMK (66.7%), GEN (23.3%), AMP (20.0%), TCY (20.0%), TOB (13.3%), SXT (13.3%), and NAL (10.0%). It was noted that one isolate displayed resistance to seven antimicrobials (AMP, AMK, GEN, STR, TOB, TCY, NAL, and SXT). Approximately 93% (28 isolates) of the isolates tested were resistant to at least one antimicrobial agent (Table 1). Non-susceptibility to STR was the most common at 100%, followed by non-susceptibility to AMK (96.7%), GEN (80.0%), TOB (80.0%), and AMP (73.3%). While three isolates exhibited non-susceptibility to nine antimicrobials, six isolates (20%) revealed MDR (Table 1). All isolates obtained from WWTPs were susceptible to CHL.

The prevalence of AMR in the isolates obtained from drainage water from areas of different land use (crop, forest, pasture, and urban) is shown in Figure 4A–E. Among 30 isolates obtained from crop land (Figure 4A), resistance to STR was the most common at 70.0%, followed by resistance to AMK (66.7%), AMP (23.3%), GEN (20.0%), and TOB (16.7%). Approximately 93% (28 isolates) of the isolates were resistant to at least one antimicrobial agent (Table 1). Non-susceptibility to STR was the most common at 100%, followed by non-susceptibility to AMK (96.7%), TOB (86.7%), GEN (73.3%), AMP (66.6%), and AMC (16.7%). None of the isolates displayed MDR (Table 2). All isolates obtained from cropland were susceptible to NAL and SXT.

Out of 30 isolates obtained from forestland (Figure 4B), resistance to STR was the most common at 86.7%, followed by resistance to AMK (76.7%), AMP (33.3%), and TOB (26.7%). Approximately 93% (28 isolates) of the isolates were resistant to at least one antimicrobial agent (Table 1). Non-susceptibility to AMK and STR was the most common at 100%, followed by non-susceptibility to GAMP (93.3%), GEN (86.7%), TOB (83.4%), AMC (23.3%), and MEM (23.3%). Although each one isolate was resistant and non-susceptible to five and eight antimicrobials, respectively, none of the isolates displayed MDR. All isolates obtained from forestland were susceptible to SXT.

Out of 30 isolates obtained from pastureland (Figure 4C), resistance to STR was the most common at 93.3%, followed by resistance to AMK (60.0%), AMP (36.7%), and TOB (13.3%). Approximately 97% (29 isolates) of the isolates were resistant to at least one antimicrobial agent (Table 1). Non-susceptibility to STR and TOB was the most common at 100%, followed by non-susceptibility to AMP (93.3%), AMK (93.3%), GEN (83.3%), AMC (33.3%), and MEM (23.4%). All isolates obtained from pastureland were susceptible to SXT.

Among 30 isolates obtained from urban land (Figure 4D), resistance to STR was the most common at 86.7%, followed by resistance to AMK (70.0%), AMP (43.3%), and TOB (33.3%). Approximately 97% (29 isolates) of the isolates were resistant to at least one antimicrobial agent (Table 1). Non-susceptibility to STR was the most common at 100%, followed by non-susceptibility to AMK (96.7%), TOB (96.7%), AMP (83.3%), and AMC (40.0%). Although four and three isolates were resistant and non-susceptible to four and seven antimicrobials, respectively, one isolate only exhibited MDR. All isolates were susceptible to CIP, CHL, and SXT.

Overall, out of the 120 isolates obtained from drainage water from areas of different land use (Figure 4E), resistance to STR was the most common at 84.2%, followed by resistance to AMK (68.3%), AMP (34.2%), TOB (22.5%), and GEN (12.5%). Most (114, 95%) of the isolates were resistant to at least one antimicrobial agent. Non-susceptibility to STR was the most common at 84.2%, followed by non-susceptibility to AMK (68.3%), AMP (34.2%), TOB (22.5%), and GEN (12.5%). The prevalence of AMR in the isolates obtained from the different water samples demonstrated a similar pattern, with high resistance to STR (≥70.0%) and AMK (≥60.0%), high non-susceptibility to GEN (≥53.3%) and TOB (≥63.3%), and high susceptibility to SXT (100%). However, it was noted that the isolates obtained from cropland were additionally susceptible to NAL, while the isolates from urban land were susceptible to CIP and CHL as well. MDR was exhibited in one isolate (0.8%) obtained from urban land (Table 1). More importantly, none of the isolates obtained from drainage water was susceptible to all antimicrobials tested in this study.

Out of the total 450 isolates (Figure 5) obtained in this study, resistance to STR was the most common at 73.1%, followed by resistance to AMK (54.2%), AMP (22.4%), TOB (13.1%), TCY (11.1%), and GEN (10.9%). Approximately 88% (395 isolates) of the isolates were resistant to at least one antimicrobial agent. Non-susceptibility to STR was the most common at 99.5%, followed by non-susceptibility to AMK (88.2%), TOB (77.8%), GEN (72.9%), AMP (69.1%), MEM (18.7%), AMC (18.5%), and TCY (12.9%). MDR was exhibited in 14 isolates (3.1%). Although the most effective antimicrobial was CHL, showing 97.6% susceptibility, none of the antimicrobials tested in this study was completely effective. While the antimicrobial agents commonly prescribed in the U.S. [19] are AMP (192 prescriptions per 1000 persons) and AMC (171 prescriptions per 1000 persons), a summarized report in the 2018 Virginia State and Regional Cumulative Antibiogram [20], which provided the percentage of bacterial isolates susceptible to antimicrobial agents within a healthcare facility, indicated that the *E. coli* isolates obtained from inpatients were most susceptible to AMK and MEM and non-susceptible to AMP and SXT.

The occurrence differences in AMR in the *E. coli* isolates associated with sample type and source are presented in Table 2. As for the sample type, there was no significant difference (*p* < 0.05) in the occurrence of AMR between the isolates obtained from WWTPs (3.97 ± 1.66) and those collected from drainage water in areas of different land uses (3.72 ± 0.98). Likewise, Burch et al. [21] indicated the co-occurrence of AMR genes in groundwater with human fecal sources at similar rates. Consistent with the high occurrence of AMR in the bacteria isolated from water in areas of multiple land use and from WWTPs, several scientists [6,22,23,24,25] indicated that surface waters receive WWTP effluents as well as run-off from manure-fertilized fields and animal feeding operations containing antibiotic resistance genes and serve as a central hub for the transport and dissemination of AM-resistant bacteria. In addition, Larsson and Flach [26] indicated that WWTP environments harbor dense, complex bacterial communities and may be more likely to be a spawning ground for AMR evolution than the recipient waterways.

The findings from the current study also revealed that the occurrence of AMR in the isolates obtained from WWTPs was significantly (*p* < 0.05) higher than that in the isolates from wild avian (3.41 ± 0.91) and livestock feces (3.10 ± 1.37), in a descending order. Ten and twenty percent of the isolates obtained from WWTPs revealed the highest non-susceptibility to nine antibiotics and MDR, respectively. These findings may manifest that *E. coli* subjected to WWTPs, a potential source of many antibiotics flushed via household septic systems, develop non-susceptibility and resistance to many antibiotics (multi-drug non-susceptibility and MDR, respectively). In fact, researchers [6,27,28] described sewage and WWTPs as hotspots for the dissemination of antibiotics resistance, indicating that those places constitute a particularly nutrient-rich environment, which can support high concentrations of bacteria, and are steadily seeded with sub-minimum inhibition concentrations of antibiotics. In addition, studies related agricultural land uses to the presence of TCY-resistant *E. coli* in rivers [25], and land uses associated with human fecal contamination (i.e., municipal WWTPs) were related to the presence of AM-resistant bacteria in groundwater and rivers [29,30,31].

For the comparison of the livestock sample sources, the occurrence of AMR in the isolates obtained from horse feces (4.07 ± 1.02) was the highest, with AMR occurrence also found in the isolates from pig, sheep, goat, turkey, chicken, and cattle feces in descending order. For wild avian species, the isolates obtained from seagull feces (3.77 ± 0.80) showed a significantly (*p* < 0.05) higher AMR than those from goose feces (3.18 ± 0.91). Studies implicated [6,32,33,34] birds, especially water-feeding birds including seagulls, in the spread of AM-resistant bacteria, presumably contracted via contact with contaminated water. Our findings showed no significant (*p* > 0.05) difference in the occurrence of AMR in the isolates obtained from water from drainage areas of different land uses. In the current study, the overall occurrence of AMR in the isolates obtained from WWTPs was the highest (3.97 ± 1.66), and that in the isolates from cattle feces was the lowest (2.32 ± 1.05). 

With regard to the variability in the prevalence and diversity of AMR in bacteria, we hypothesize that it is possible for *E. coli* isolates to develop different degrees of resistance to antimicrobials depending upon the type of environment to which they have been exposed and their genetic background. Doyle et al. [35] also indicated that this variability might be due to substantive differences in antimicrobial usage and practices. While Hernando-Amado et al. [36] reported that contributing factors to the incidence of AMR in bacteria span the One Health spectrum, including human, agricultural, and environmental dimensions, several scientists [37,38,39,40] reported that the antimicrobial use in human medicine and livestock production contributes to the emergence of bacteria carrying AMR, and AM-resistant bacteria excreted in feces can be transported through environmental compartments, including soil, water, and air [21]. In addition, a CDC report [41] indicated that AMR germs and genes that cause resistance traits are present in the environment and can spread in waterways and soils. The report also cited measurable levels of resistant bacteria in surface waters. However, scientists do not fully understand the risk posed by antimicrobial resistance in the environment for human health. 

Therefore, understanding environmental factors that modulate microbial fate and transport is important for formulating useful policy and management interventions to reduce human exposure via environmental routes [40]. In addition, water connects physically distinct landscapes, serving to transport AM-resistant bacteria within and across landscapes [21]. This is concerning, because our data (Table 2) revealed the high occurrence of AMR in bacteria obtained from surface water, from areas of all land uses. Likewise, studies [40,41,42,43,44,45,46] highlighted groundwater as a global reservoir of AMR in bacteria.

Regardless, this study describes the first documented research identifying potential hotspots of AMR emergence and dissemination in the environment. The findings will be helpful for policymakers and researchers to identify gaps in knowledge about the links between AMR and the environment. Continued research efforts on a larger scale are needed to confirm the environmental impact on the prevalence variance of AMR in bacteria.

Furthermore, this study simply indicates the occurrence of AMR in *E. coli* in randomly selected environmental samples from Central Virginia. Due to the limited availability of the same livestock and avian species in different farms and environments, respectively, each acquired sample may not represent all environmental samples in the study area. However, the findings are noteworthy to understand *E. coli* ecology in our environment, identifying potential hotspots of AMR emergence and dissemination, and may contribute to the effective development of mitigation strategies for AMR in bacteria. The authors would like to declare that this study was carried out, mainly for academic research purposes, without any conflicts of interest.

## 3. Materials and Methods

### 3.1. Sample Collection

Table 3 presents the coordinates of the sampling locations and the sources of the samples examined in this study. Due to the different amounts of feces produced by the farm livestock (cattle, chicken, goat, horse, pig, sheep, and turkey) and wild avian species (goose and seagull), approximately ten and two grams of fecal samples were collected from the upper layer of feces on the ground, respectively, ensuring no contact with animals and other materials. Additionally, 500 mL water samples were obtained from wastewater treatment plants (WWTPs) and drainage areas of different land use categories, including crop fields, forests, pastures, and urban areas throughout Central Virginia, between August 2020 and February 2021. A total of 29 locations consisting of 14 livestock farms, 5 wild environments associated with avians, 8 drainage areas, and 2 WWTPs were studied. 

### 3.2. E. coli Isolation 

Fresh samples aseptically collected from each sampling location were transported to our laboratory in insulated containers packed with ice. All samples were kept in the refrigerator (4 ± 2 °C) and used for *E. coli* isolation within 24 h after arrival. For the fecal samples of livestock and wild avians, one gram of the homogenized samples was stomached with 2 mL of phosphate-buffered saline (PBS, unless otherwise stated; all media were from Bacto, BD, Sparks, MD, USA) solution in a laboratory blender (Model 400 Circulator, Seward Ltd., West Sussex, UK) at 260 rpm for 2 min. One-tenth of the homogenate was surface-plated onto a modified mTEC agar plate [47] and incubated at 44.5 ± 0.2 °C for 22 h. Three *E. coli* isolates from each of five different samples per sampling location (a total of 15 *E. coli* isolates per site) were attempted to obtain. For the water samples, five *E. coli* isolates were obtained from each of three different samples per sampling location (a total of 15 *E. coli* isolates per site). The isolation of *E. coli* was conducted by following the EPA Method 1603 [47]. In brief, a 500 mL water sample was filtered through a membrane (0.45 µm, Millipore Sigma, Billerica, MA, USA). The membrane containing the bacteria was placed on an mTEC agar plate, incubated at 35 ± 0.5 °C for 2 h to resuscitate the injured or stressed bacteria, and then incubated at 44.5 ± 0.2 °C for 22 h. 

The colonies of assumptive *E. coli* on mTEC agar showing a red or magenta color after the incubation period were randomly selected and subjected to the API 20E biochemical test (bioMe’rieux, Hazelwood, MO, USA) for confirmation by following the manufacturer’s instruction. All confirmed *E. coli* isolates were suspended in Brucella broth containing 20% glycerol and stored at −80 °C until used for further evaluation of antimicrobial resistance (AMR). 

### 3.3. AMR Test

Following the procedure described by Kim et al. [48], antimicrobial susceptibility tests were performed on Mueller–Hinton Agar (MHA) using the Kirby–Bauer disk diffusion method [14]. In brief, the confirmed *E. coli* isolates were tested for susceptibility to 12 antimicrobial agents approved by the U.S. Food and Drug Administration for clinical use, and their categories are shown in Table 4. Antimicrobial susceptibility, classified as “susceptible”, “intermediate”, and “resistant”, was interpreted in accordance with criteria established by the National Committee of Clinical Laboratory Standards [14]. In addition, bacteria classified as either resistant or intermediate were defined as “non-susceptible”, and those exhibiting resistance to at least one antimicrobial agent in three or more antimicrobial categories were defined as multi-drug resistant (MDR) [49,50]. *E. coli* ATCC 25922 was used as a control strain for the performance of the antimicrobials used in this study.

One loop of each confirmed *E. coli* isolate was transferred to 10 mL of Mueller–Hinton broth (MHB) and incubated at 36 °C for 24 h. The isolates were again sub-cultured in MHB to ensure that they were all viable and fresh before AMR testing. *E. coli* ATCC 25922 was also cultured and sub-cultured similarly in MHB. One-tenth of each *E. coli* suspension in MHB, adjusted to approximately 8 log CFU/mL, was transferred onto MHA plates and spread uniformly. Before applying the antimicrobial discs, the plates were left for 10 min to allow any excess surface moisture to be absorbed. Then, antimicrobial discs were placed on the plates by using the Thermo Scientific Remel Antimicrobial Susceptibility 12-place 150 mm Disk Dispenser (ST1215, Oxoid Ltd., Basingsoke, UK). The plates were incubated for 24 h at 36 °C, and the inhibition diameter zones were measured in millimeters in each plate with a caliper and recorded for each sample. 

### 3.4. Data Analysis

For statistical analysis purposes, values of 0, 0.5, and 1 were assigned to *E. coli* classified as susceptible, intermediate, and resistant, respectively, with respect to each of the 12 antimicrobials tested. The data were analyzed using the general linear model procedures. Comparisons of the means were performed using Duncan’s multiple range test (SAS Institute, Cary, NC, USA) to determine the significance of the differences (*p* < 0.05).

## 4. Conclusions

This study demonstrates the risk of different environments as natural habitats for AM-resistant *E. coli*. While the nature of the bacteria associated with the environmental samples studied and their specific AMR prevalence are still unknown, the foregoing analysis suggests that the inappropriate use and misuse of antibiotics result in the contamination of various aspects of the environment. In addition, while water from WWTPs and areas of various land uses appears to be a hotspot for the emergence and spread of AMR, considering AMR prevalence in other tested sources, adopting One Health mitigation strategies that address human, animal, and environmental AMR sources may prove more effective than targeting a singular source. Furthermore, this research has national relevance, portraying a scalable AMR research model to understand *E. coli* ecology in our environment. Further continued research efforts are essential on a larger scale to validate the environmental influence on the observed difference in AMR prevalence and the associated antibiotic resistance genes in bacteria. 

## Figures and Tables

**Figure 1 antibiotics-13-00223-f001:**
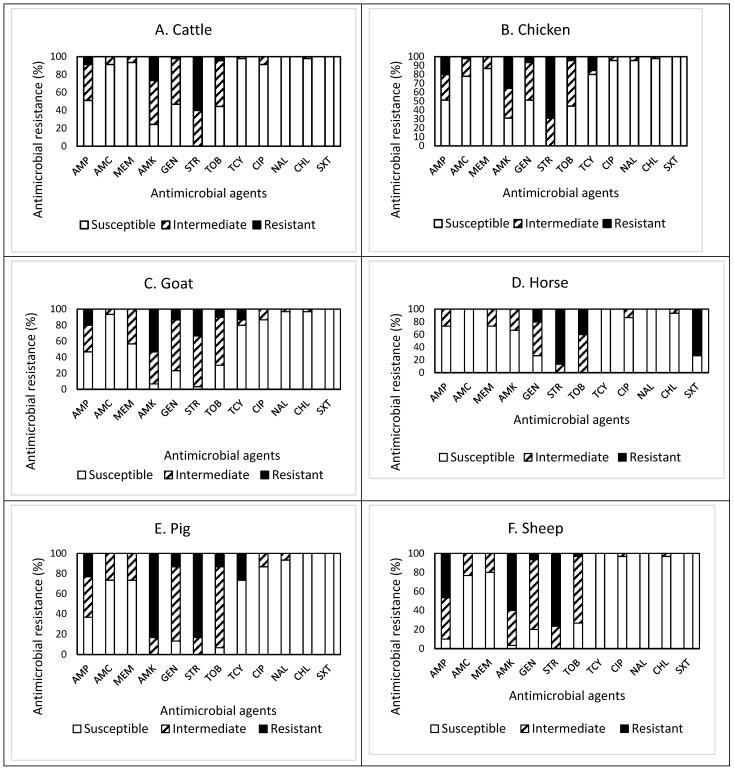
(**A**–**H**) Prevalence of resistance to 12 antimicrobial agents in a total of 225 *E. coli* isolates obtained from livestock: cattle ((**A**), n = 45), chicken ((**B**), n = 45), goat ((**C**), n = 30), horse ((**D**), n = 15), pig ((**E**), n = 30), sheep ((**F**), n = 30), turkey ((**G**), n = 30), and overall livestock ((**H**), n = 225).

**Figure 2 antibiotics-13-00223-f002:**
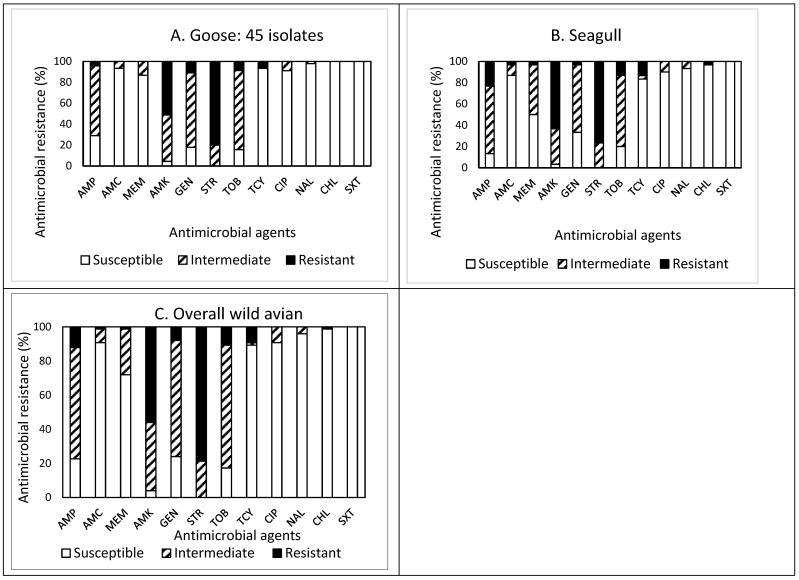
(**A**–**C**) Prevalence of resistance to 12 antimicrobial agents in a total of 75 *E. coli* isolates obtained from wild avian species: goose ((**A**), n = 45), seagull ((**B**), n = 30), and overall wild avians ((**C**), n = 75).

**Figure 3 antibiotics-13-00223-f003:**
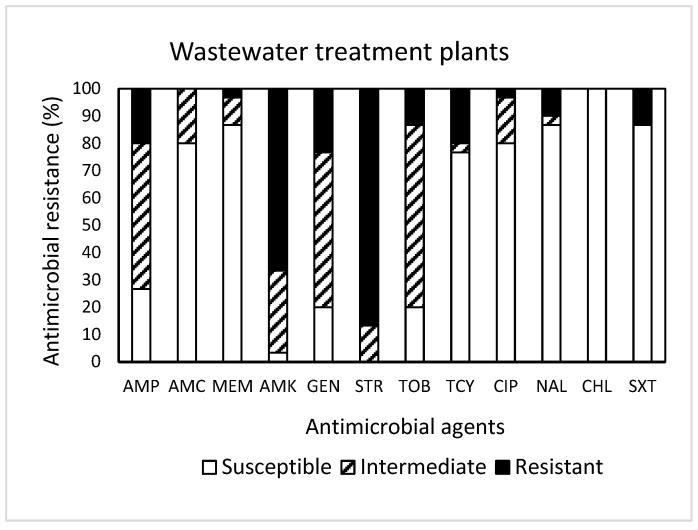
Prevalence of resistance to 12 antimicrobial agents in a total of 30 *E. coli* isolates obtained from wastewater treatment plants.

**Figure 4 antibiotics-13-00223-f004:**
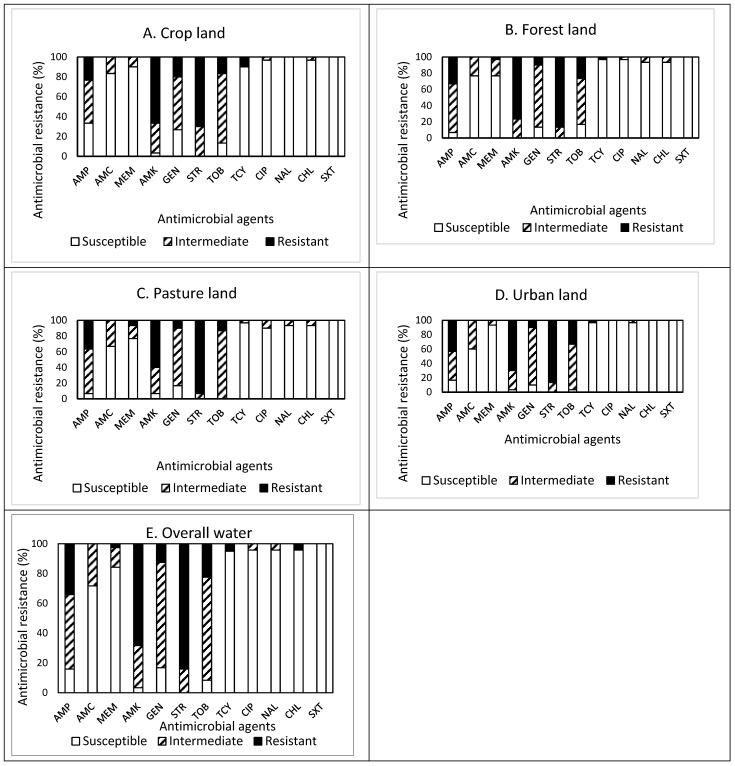
(**A**–**E**) Prevalence of resistance to 12 antimicrobial agents in a total of 120 *E. coli* isolates obtained from drainage water in areas associated with different land use: crop land ((**A**), n = 30), forest land ((**B**), n = 30), pasture land ((**C**), n = 30), urban land ((**D**), n = 30), and overall water ((**E**), n = 120).

**Figure 5 antibiotics-13-00223-f005:**
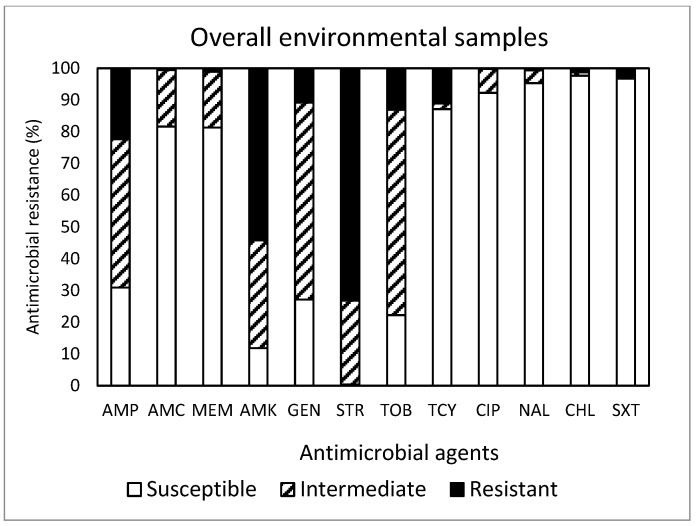
Prevalence of resistance to 12 antimicrobial agents in a total of 450 *E. coli* isolates obtained from the environmental samples tested in this study.

**Table 1 antibiotics-13-00223-t001:** Prevalence (%) of resistance and non-susceptibility to one or more antimicrobial agents for 450 *E. coli* isolates obtained from environmental samples in Central Virginia between August 2020 and February 2021 *.

Sample			Nature of AMR ^b^	Quantity of Antimicrobial Agents ^c^									
Type	Source (n ^a^)	Location (n)		1	2	3	4	5	6	7	8	9	MDR ^d^ (≥3)
Farm animal /livestock	Cattle (45)	3	R	46.7	17.8	6.7	0.0	0.0	0.0	0.0	0.0	0.0	0.0
			R + I	15.6	4.4	22.2	28.9	20.0	6.7	2.2	0.0	0.0	NA ^e^
	Chicken (45)	3	R	37.8	33.3	6.7	4.4	2.2	0	0	0.0	0.0	2.2
			R + I	17.8	15.6	17.8	6.7	13.3	17.8	6.7	0.0	4.4	NA
	Goat (30)	3	R	13.3	26.7	26.7	6.7	0.0	0.0	0.0	0.0	0.0	0.0
			R + I	3.3	13.3	10.0	10.0	26.7	16.7	0.0	6.7	0.0	NA
	Horse (15)	1	R	20.0	13.3	26.7	40.0	0.0	0.0	0.0	0.0	0.0	0.0
			R + I	0.0	0.0	6.7	13.3	40.0	33.3	0.0	6.7	0.0	NA
	Pig (30)	2	R	6.7	53.3	23.3	10.0	0.0	3.3	0.0	0.0	0.0	10.0
			R + I	0.0	3.3	0.0	23.3	30.0	16.7	20.0	6.7	0.0	NA
	Sheep (30)	1	R	20.0	40.0	26.7	3.3	0.0	0.0	0.0	0.0	0.0	0
			R + I	0.0	0.0	6.7	30.0	40.0	13.3	10.0	0.0	0.0	NA
	Turkey (30)	1	R	26.7	30.0	10.0	10.0	3.3	0.0	0.0	0.0	0.0	6.7
			R + I	6.7	10.0	13.3	23.3	16.7	23.3	6.7	0.0	0.0	NA
	Total (225)	14	R	27.1	31.1	16.0	7.6	0.9	0.4	0.0	0.0	0.0	2.7
			R + I	8.0	7.6	12.4	19.6	24.4	16.4	8.4	2.2	0.9	NA
Wild avian	Goose (45)	3	R	33.3	28.9	17.8	4.4	0.0	0.0	0.0	0.0	0.0	0.0
			R + I	0.0	4.4	6.7	26.7	42.2	15.6	2.2	2.2	0.0	NA
	Seagull (30)	2	R	10.0	53.3	23.3	6.7	0.0	0.0	0.0	0.0	0.0	3.3
			R + I	0.0	0.0	3.3	10.0	50.0	16.7	20.0	0.0	0.0	NA
	Total (75)	5	R	24.0	38.7	20.0	5.3	0.0	0.0	0.0	0.0	0.0	1.3
			R + I	0.0	2.7	5.3	20	45.3	16.0	9.3	1.3	0.0	NA
Land use	Crop land (30)	2	R	33.3	20	26.7	13.3	0.0	0.0	0.0	0.0	0.0	0.0
			R + I	0.0	6.7	10.0	26.7	30.0	16.7	10.0	0.0	0.0	NA
	Forest land (30)	2	R	10.0	40.0	26.7	13.3	3.3	0.0	0.0	0.0	0.0	0.0
			R + I	0.0	0.0	3.3	20.0	33.3	33.3	6.7	3.3	0.0	NA
	Pasture land (30)	2	R	16.7	43.3	30.0	6.7	0.0	0.0	0.0	0.0	0.0	0.0
			R + I	0.0	0.0	0.0	13.3	33.3	33.3	10.0	6.7	0.0	NA
	Urban land (60)	2	R	10.0	36.7	36.7	13.3	0.0	0.0	0.0	0.0	0.0	3.3
			R + I	0.0	0.0	0.0	23.3	43.3	23.3	10	1.7	0.0	NA
	Total (120)	8	R	17.5	35.0	30.0	11.7	0.8	0.0	0.0	0.0	0.0	0.8
			R + I	0.0	1.7	4.2	20.8	35.0	26.7	9.2	2.5	0.0	NA
Wastewater treatment plant (30)		2	R	16.7	33.3	23.3	6.7	3.3	6.7	3.3	0.0	0.0	20.0
			R + I	3.3	0.0	3.3	30.0	20.0	23.3	10.0	0.0	10.0	NA
Overall (450)		29	R	23.3	33.6	20.9	8.2	0.9	0.7	0.2	0.0	0.0	3.1
			R + I	4.2	4.7	8.4	20.7	30.4	19.6	8.9	2.0	1.1	NA

* Susceptibility categorization was carried out in accordance with the interpretive criteria provided by the National Committee of Clinical Laboratory Standards recommendations [14]; ^a^ number of isolates tested; ^b^ antimicrobial resistance (AMR); R: resistant; I: intermediate; R + I: non-susceptible to the antimicrobial agents tested; ^c^ prevalence (%) is presented as resistance and non-susceptibility of isolates to the total number of antimicrobial agents tested [i.e., an isolate exhibiting resistance and intermediate susceptibility to two and four antimicrobial agents, respectively, is presented under column 2 for resistance and under column 6 for non-susceptibility (R + I)]; ^d^ multi-drug resistance; ^e^ not applicable.

**Table 2 antibiotics-13-00223-t002:** Occurrence differences in antimicrobial resistance in a total of 450 *E. coli* isolates from environmental samples.

Sample		Number of Sampling Location	Occurrence of Antimicrobial Resistance ^d^		
Type (n ^a^)	Source (n)				
Farm animal/livestock (225)	Cattle (45)	3	3.10 ± 1.37 C *	2.32 ± 1.05 f *	2.32 ± 1.05 D **
	Chicken (45)	3		2.71 ± 1.57 ef	2.71 ± 1.57 CD
	Goat (30) ^#^	3		3.27 ± 1.49 bcde	3.27 ± 1.49 BC
	Horse (15) ^#^	1		4.07 ± 1.02 a	4.07 ± 1.02 A
	Pig (30)	2		3.93 ± 1.09 ab	3.93 ± 1.09 AB
	Sheep (30) ^ǂ^	1		3.42 ± 0.86 abcd	3.42 ± 0.86 ABC
	Turkey (30) ^ǂ^	1		3.02 ± 0.40 de	3.02 ± 1.40 C
Wild avian (75)	Goose (45)	3	3.41 ± 0.91 BC	3.18 ± 0.91 cde	3.18 ± 0.91 B
	Seagull (30)	2		3.77 ± 0.80 abc	3.77 ± 0.80 A
Water (120) ^b∆^	Crop land (30)	2	3.72 ± 0.98 AB	3.38 ± 1.13 abcd	3.38 ± 1.13 A
	Forest land (30)	2		3.85 ± 1.06 abc	3.85 ± 1.06 A
	Pasture land (30)	2		3.82 ± 0.90 abc	3.82 ± 0.90 A
	Urban land (30)	2		3.83 ± 0.76 abc	3.83 ± 0.76 A
Waste (30) ^c∆^	Treatment plant (30)	2	3.97 ± 1.66 A	3.97 ± 1.66 a	

^a^ Number of isolates tested; ^b^ water associated with areas of different land use (crop, forest, pasture, and urban); ^c^ water associated with a wastewater treatment plant; ^d^ values of 0, 0.5, and 1 were assigned to bacteria demonstrating susceptibility, intermediate susceptibility, and resistance, respectively, to each of the 12 tested antimicrobials, for statistical analysis purposes; ^#^ due to the lower occurrence of *E. coli* than expected in some of the samples collected, we were unsuccessful in obtaining the target number of isolates (15 isolates per sampling location); ^ǂ^ for farms with a limited availability of the same livestock (i.e., sheep and turkey), three *E. coli* isolates from ten different samples (up to 30 *E. coli* isolates) were obtained; ^∆^ for water and wastewater treatment plants, five *E. coli* isolates from each of three different samples per site was obtained; * in the same column, means followed by the same uppercase and lowercase letter, respectively, are not significantly different (*p* > 0.05); ** means followed by the same uppercase letter in the same column within the same sample type are not significantly different (*p* > 0.05).

**Table 3 antibiotics-13-00223-t003:** GPS coordinates of the samples collected.

Sample			GPS Coordinates in Degrees, Minutes, and Seconds
Type	Source (n ^a^)	Location (n)	
Farm animal /livestock	Cattle (45)	3	36°54′6.084″ N, 76°44′51.54″ W
			37°9′53.2008″ N, 77°24′38.2176″ W
			37°16′6.8988″ N, 77°27′57.744″ W
	Chicken (45)	3	37°13′56.3484″ N, 77°26′37.0284″ W
			37°4′59.664″ N, 76°43′26.076″ W
			37°15′5.76″ N, 77°18′1.224″ W
	Goat (30)	3	37°13′51.7692″ N, 77°26′52.4472″ W
			37°4′59.664″ N, 76°43′26.076 W
			37°15′5.76″ N, 77°18′1.224″ W
	Horse (15)	1	37°11′26.7252″ N, 77°25′27.2496″ W
	Pig (30)	2	37°37′34.932″ N, 76°53′35.412″ W
			37°4′59.664″ N, 76°43′26.076″ W
	Sheep (30)	1	37°13′58.2384″ N, 77°25′53.922″ W
	Turkey (30)	1	37°37′34.932″ N, 76°53′35.412″ W
Wild avian	Goose (45)	3	37°14′25.44″ N, 77°25′14.88″ W
			37°16′6.8988″ N, 77°27′57.744″ W
			37°13′43.8672″ N, 77°26′15.414″ W
	Seagull (30)	2	37°9′53.2836″ N, 77°22′3.7272″ W
			37°13′43.8672″ N, 77°26′15.414″ W
Land use	Crop land (30)	2	37°13′28.6″ N, 77°26′46.7″W
			37°14′04.2″ N, 77°26′26.4″W
	Forest land (30)	2	37°24′59.472″ N, 78°38′10.6872″ W
			37°15′28.1592″ N, 78°29′10.2768″ W
	Pasture land (30)	2	37°13′07.0″ N, 77°22′29.1″W
			37°20′2.68″ N, 77°13′48.35″W
	Urban land (60)	2	37°13′9.2424″ N, 77°24′59.508″ W
			37°13′12.5436″ N, 77°24′54.8568″ W
Wastewater treatment plant (30)		2	37°14′19.968″ N, 77°23′35.52″ W
			37°17′42″ N, 77°15′32.4″ W

^a^ Number of isolates obtained.

**Table 4 antibiotics-13-00223-t004:** A list of antimicrobials and the interpretive criteria used in this study [14,48] *.

Antimicrobial Category	Antimicrobial Agent and Its Abbreviation	Concentration (µg/Disk)	Zone Diameter (mm)		
			S	I	R
Penicillins	Ampicillin (AMP)	10	>17	14–16	<13
β-lactamase inhibitor combinations	Amoxicillin–clavulanic acid (AMC)	30	>18	14–17	<13
Carbapenems	Meropenem (MEM)	10	>23	20–22	<19
Aminoglycosides	Amikacin (AMK)	30	>17	15–16	<14
	Gentamicin (GEN)	10	>15	13–14	<12
	Streptomycin (STR)	10	>15	12–14	<11
	Tobramycin (TOB)	10	>15	13–14	<12
Tetracyclines	Tetracycline (TCY)	30	>15	12–14	<11
Fluoroquinolones	Ciprofloxacin (CIP)	5	>21	16–20	<15
Quinolones	Nalidixic acid (NAL)	30	>19	14–18	<13
Phenicols	Chloramphenicol (CHL)	30	>18	13–17	<12
Folate pathway inhibitors	Trimethoprim–sulfamethoxazole (SXT)	25	>16	11–15	<10

* Interpretive criteria: S, susceptible; I, intermediate; and R, resistant to the antimicrobial agents tested.

## Data Availability

Data are contained within the article.

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
