# Peer review of "Examining Antimicrobial Resistance in Escherichia coli: A Case Study in Central Virginia’s Environment"

_antibiotics, 2024, doi:10.3390/antibiotics13030223_

Round 1

Reviewer 1 Report

Comments and Suggestions for Authors

Title: A case study on the prevalence variance of antimicrobial resistance in E. coli associated with environment in Central Virginia

This manuscript is well-written by the authors. I do believe that if they can improve the manuscripts following all comments. It might have a chance to publish in the journal.

Comments

1. Topic: Please rewrite. It is complicated. Please write the full name of Escherichia coli.

2. Line 7-9: Please re-write. Please write the full address of each no. (1, 2, 3).

Abstract
3. The first sentence in the abstract should be re-write.

4. Line 19: Please write the full name of Escherichia coli.

5. Line 31 and 33: E. coli should be written in italic.

6. Why does the author focus on E. coli? Please give the reason or describe in the abstract.

Introduction

7. Line 37, 90: Please remove the underline.

8. I strongly suggest the authors to modify or re-write the introduction. What distinguishes your research between other? please give state of the art of your research

            -The first paragraph: describe the importance of antimicrobial resistance in human, animals, as well as environment. Try to combine the information as one health.

            -The second paragraph: describe the importance of E. coli that used as the bio-monitor in this study. Why does the author focus on E. coli?

-The third paragraph: describe the importance of Central Virginia. Why does the author focus on this area?

            -The fourth paragraph: describe the objective of this study. Why the authors are interested in this study?

Methodology

9. Do the authors perform a document on ethics? If not, please describe about the sample collection. How did the authors collect the samples? For example, the samples were collected from the farms…

10. Line 118, 121, 127, 135: Please edit the writing of the word "E. coli".

11. Table 2 should be presented as supplementary data.

 Results and Discussion

12. Figures: It would be better if the authors prepare the figures using the same type of the alphabet as the text.

13. The main results of this study are antibiotic susceptibility test based on disc diffusion assay. Then, the authors presented raw data of the antibiotic susceptibility as Fig. 1 - Fig. 5, and Table 4. It would be better if the author added some experiments to achieve results.

14. I recommend detection of antibiotic resistance genes that are involved in the resistance to the antibiotics.

15. Please add the information of the discussion. Try to compare the results (the author’s hypothesis) with other finding by other researchers.

Conclusion

16. Please modify the conclusion. It should be summarized on the key finding. Please don’t repeat the results. Importantly, the authors should not add the references in the conclusion.

References

17. The references of 2022,2021, and 2023 are suggested to be cited.

18. Please remove some old references.

19. Actually, 30-40 references are enough for the research article. Please delete some unnecessary references or old references. Please edit the references.

Comments on the Quality of English Language

-

Author Response

This manuscript is well-written by the authors. I do believe that if they can improve the manuscripts following all comments. It might have a chance to publish in the journal.

Comments

  1. Topic: Please rewrite. It is complicated. Please write the full name of Escherichia coli.

The topic and E. coli were rewritten as suggested by the reviewer.

  1. Line 7-9: Please re-write. Please write the full address of each no. (1, 2, 3).

The full addresses of each 1, 2, and 3 are included.

Abstract
3.    The first sentence in the abstract should be re-write.

As suggested by the reviewer, to improve the quality of the manuscript, the first sentence was removed.

  1. Line 19: Please write the full name of Escherichia coli.

The initial mention of E. coli was revised to its full name, Escherichia coli, as suggested by the reviewer.

  1. Line 31 and 33:  colishould be written in italic.

They were italicized in the manuscript, as suggested by the reviewer.

  1. Why does the author focus on  coli? Please give the reason or describe in the abstract.

The rationale was described in the second sentence of the abstract and introduction section, and the contents were revised accordingly.

Introduction

  1. Line 37, 90: Please remove the underline.

Lines 37 and 90 in the manuscript submitted to the Antibiotics journal do not have underlines. They may have been added to the submitted manuscript due to compatibility issues with the journal's software.

  1. I strongly suggest the authors to modify or re-write the introduction. What distinguishes your research between other? please give state of the art of your research

-The first paragraph: describe the importance of antimicrobial resistance in human, animals, as well as environment. Try to combine the information as one health.

The first paragraph was rewritten as suggested by the reviewer.

-The second paragraph: describe the importance of E. coli that used as the bio-monitor in this study. Why does the author focus on E. coli?

The second paragraph was rewritten as suggested by the reviewer.

-The third paragraph: describe the importance of Central Virginia. Why does the author focus on this area?

The third and fourth paragraphs were rewritten as suggested by the reviewer and combined together.

-The fourth paragraph: describe the objective of this study. Why the authors are interested in this study?

The fourth paragraphs were re-written as suggested by the reviewer and combined with the third paragraph.

Methodology

  1. Do the authors perform a document on ethics? If not, please describe about the sample collection. How did the authors collect the samples? For example, the samples were collected from the farms…

Fecal samples were gathered from the ground in farms and wild avian settings without direct contact with the animals. The sentence was revised as suggested by the reviewer.

  1. Line 118, 121, 127, 135: Please edit the writing of the word "E. coli".

The format for the word "E. coli" may have been changed due to compatibility issues with the journal's software.

  1. Table 2 should be presented as supplementary data.

The authors maintained the integrity of Table 2 in its current form but remain open to presenting it as supplementary data if advised to do so by the journal. 

Results and Discussion

  1. Figures: It would be better if the authors prepare the figures using the same type of the alphabet as the text.

The authors presented the figures using the same type of alphabet as the text. They may look different due to compatibility issues with the journal's software.

  1. The main results of this study are antibiotic susceptibility test based on disc diffusion assay. Then, the authors presented raw data of the antibiotic susceptibility as Fig. 1 - Fig. 5, and Table 4. It would be better if the author added some experiments to achieve results.

The methodology used to achieve results shown in Figures 1-5 and Table 4 are illustrated in Lines 184-192 of the “Methodology” section.

  1. I recommend detection of antibiotic resistance genes that are involved in the resistance to the antibiotics.

The data provided in this manuscript offers intriguing yet insights on a relatively modest scale. The authors are actively expanding their bacterial isolate collection on a larger scale to validate the environmental influence on the observed variations in AMR prevalence among E. coli. Upon acquiring a more substantial sample size, the authors intend to explore antibiotic resistance genes associated with resistance patterns in conjunction with bacterial cultures obtained in previous studies. The reviewer’s recommendation falls outside the current scope of the project.

  1. Please add the information of the discussion. Try to compare the results (the author’s hypothesis) with other finding by other researchers.

Given the nature of the work presented, the authors have opted to combine the results and discussion sections into a unified "Result and Discussion." A comprehensive comparison of the authors' findings and hypotheses with those of others is illustrated and can be found in this section.

  1. Please modify the conclusion. It should be summarized on the key finding. Please don’t repeat the results. Importantly, the authors should not add the references in the conclusion.

The conclusion was modified as suggested by the reviewer.

References

  1. The references of 2022,2021, and 2023 are suggested to be cited.

Recent references have been incorporated as suggested by the reviewer. Nevertheless, to recognize seminal contributions and foundational discoveries, earlier references have also been included.

  1. Please remove some old references.

Old references are removed as suggested by the reviewer.

  1. Actually, 30-40 references are enough for the research article. Please delete some unnecessary references or old references. Please edit the references.

Outdated references have been eliminated in accordance with the reviewer's suggestions. However, recognizing the need for other pertinent references, the authors have concurred to retain additional citations.

We feel these changes have improved the manuscript and trust you will let us know if anything else is required. Thank you very much for your help.

Sincerely,

Reviewer 2 Report

Comments and Suggestions for Authors

needs to be clarified:

1. how do I choose the sampling sites?

how do I collect the samples (second method), number of samples, and the sampling procedure,

2. methodology for water and soil sampling.

3. Do you take the faeces directly from the animals or from the soil? please indicate

4. do you also need a figure showing the geographical location of the study area? 

5. do you calculate the livestock population?

6. why do you use that E.coli sepa? 

7. I believe that generalised linear mixed models (GLMM) with a logit link function may be more helpful in exploring factors associated with antimicrobial resistance. 

Author Response

needs to be clarified:

  1. how do I choose the sampling sites?

Sampling locations for farms with livestock, wild avian, wastewater treatment plants (WWTP), and water in different land use areas in central Virginia were selected randomly. Due to the restricted availability of the same species of livestock and wild avian across various farms and environments, the number of sampling locations for each source may not be uniform. Additionally, only a limited number of farms expressed interest in participating in the current study, as detailed in the first paragraph of the "Results and Discussion" section.

how do I collect the samples (second method), number of samples, and the sampling procedure,

Details can be found in the “Sample collection” and "E. coli isolation" sections under the Methodology.

  1. methodology for water and soil sampling.

Details on the methodology employed for water samples can be found in the "E. coli isolation" section under the Methodology. It is noteworthy that no soil samples were utilized; rather, only fecal and water samples were included in the study.

  1. Do you take the faeces directly from the animals or from the soil? please indicate

Samples were collected from the upper layer of feces on the ground, ensuring no contact with other materials. In response to the reviewer's suggestion, the authors incorporated this information into the "Sample collection" segment of the Methodology.

  1. do you also need a figure showing the geographical location of the study area? 

The authors maintained the original format of Table 1 to showcase relevant variables (i.e., sample type, source, E. coli isolate numbers) in connection with the GPS coordinates.

  1. do you calculate the livestock population?

Livestock population calculations were not conducted. Nevertheless, in this pilot study, fecal samples were acquired from farms with a minimum of five animals.

  1. why do you use that E.coli sepa? 

Due to their widespread occurrence in the environment and their ability to serve as reservoirs of antibiotic resistance genes, E. coli are commonly employed as indicators in numerous antimicrobial resistance monitoring programs. The level of AMR in E. coli is also considered a reliable indicator of the selection pressure exerted by antimicrobial use in humans and agricultural practices. Therefore, E. coli species were used for the study, and the information is included in the “Introduction” section.

  1. I believe that generalised linear mixed models (GLMM) with a logit link function may be more helpful in exploring factors associated with antimicrobial resistance. 

Given that the statistical analysis employed for the results involves a straightforward comparison of AMR prevalence in E. coli from samples, the authors have agreed to assign values of 0, 0.5, and 1 to indicate the levels of AMR susceptibility: susceptible, intermediate, and resistant, respectively, to the tested antimicrobials. The data were then analyzed using a generalized linear model. Therefore, the authors revised the “Data analysis” section accordingly.

We feel these changes have improved the manuscript and trust you will let us know if anything else is required. Thank you very much for your help.

Sincerely,

Round 2

Reviewer 2 Report

Comments and Suggestions for Authors

The amount of sample collected (feces, water) has not been added.

From line 156-163 place it in an appropriate size. 

Check that E.coli is written the same throughout the document, line 109-116 are different from the rest of the document. 

Author Response

The authors appreciate the thoughtful suggestions provided by the reviewer for the matter of revision. Following are our responses to the reviewer's comments. We have made most of the changes suggested by the reviewer, and a list of our itemized responses to the reviewers is addressed below.

The amount of sample collected (feces, water) has not been added.

Due to the different amounts of feces produced by the farm livestock (cattle, chicken, goat, horse, pig, sheep, and turkey) and wild avian species (goose and seagull), approximately ten and two grams of feces on the ground, respectively, were collected for the isolation of E. coli. In addition, 500 ml of water samples were collected from wastewater treatment plants (WWTP) and drainage areas of different land use categories for the isolation of E. coli. In response to the reviewer's comment, the authors incorporated this information into the "Sample collection" and "2.2 E. coli isolation" segments of the Methodology.

From line 156-163 place it in an appropriate size. 

The font size has been reduced to 9, as suggested by the reviewer.

Check that E.coli is written the same throughout the document, line 109-116 are different from the rest of the document. 

The words "E. coli" have been re-formatted to the same throughout the document, as suggested by the reviewer.

We feel that these changes have improved the manuscript and trust that you will let us know if anything else is required. Thank you very much for your suggestion.